# A Roof of Greenery, but a Sky of Unexplored Relations—Meta-Analysis of Factors and Properties That Affect Green Roof Hydrological and Thermal Performances

**Mithun Hanumesh [1,2], Rémy Claverie [1] and Geoffroy Séré [2,*]**

1   Cerema DTer-Est/LRN, 71 Rue de la Grande Haie, F-54510 Tomblaine, France;
    mithun.hanumesh@cerema.fr (M.H.); remy.claverie@cerema.fr (R.C.)
2   Laboratoire Sols et Environnement, Université de Lorraine-INRA, 2 Avenue de la Forêt de Haye, BP 172,
    F-54505 Vandoeuvre lès Nancy, France
*   Correspondence: geoffroy.sere@univ-lorraine.fr

**Abstract:** Green roofs are expected to contribute to the mitigation of multiple environmental issues that affect urban areas. Owing to their composition, organization, and external factors, the performances of green roofs have been demonstrated to be overall positive but strongly variable. Our work first aims at proposing consensual definitions and a frame adapted to these biotic-abiotic systems. It also aims at shedding light on the qualitative relations between various internal properties and external factors of green roofs on their hydrological and thermal performances. One hundred relevant study papers were filtered from 395 papers as per our defined search criteria based on originality and precision. The expectations were to be capable of hierarchizing factors and properties that would influence the performances of green roofs. The main findings highlighted that most factors and properties have a positive influence on the performances of green roofs, showing there are many existing levers to enhance the green roof performances and tackle some of the main urban environmental issues. However, even if previous research has already explored various relations, in the final filtered consideration of 6 performances and 30 factors and properties, there was a possibility of 180 combined factor–property–performance relations studies overall. Out of these possibilities, only 82 have been studied at least once, leaving the other 98 relations (54%) unexplored. Considering that these lists were far from exhaustive, a huge potential in determining green roof performances remains unearthed. In this regard, various proposals have been made regarding: (i) identification of levers to enhance the performances of green roofs; (ii) filling the gaps: the exploration of the unstudied relations; (iii) promotion of deeper and innovative experimental approaches for research on green roof performances; and (iv) the shift from mono to transdisciplinary research about green roofs.

**Keywords:** substrate; leaf area index; climate; meteorological conditions; retention; detention; thermal insulation; energy demand; ecosystem services

## 1. Introduction

The constant expansion in terms of space and density of urban areas has numerous consequences on city infrastructure dynamics and surrounding environment mainly through impervious surfaces (such as roofs, paved driveways, roads, and paved parking lots) occupying a predominant place in these spaces [1,2]. They replace the vegetated, pervious surfaces and create the danger of waterproofing cities, a phenomenon that promotes increase of surface runoff. Furthermore, fewer vegetated areas in turn cause a decrease in canopy interception and transpiration within the city, leading to increased temperature and decreased air humidity, leading to environmental issues, such as urban heat island and massive consumption and loss of energy [3].

Since the 1950s and mainly in the past decade, to address such concerns, new forms of nature-based solutions for urban areas have been emerging. However, many of those

techniques require availability of land space, which is usually not available in densely built downtown urban areas [4]. In every city, the availability of roof area is very likely. Hence, among the techniques allowing introduction of nature into the city, green roofs hold a significant potential. Integrated into the building, they do not consume any additional space and can even be implemented on existing buildings (a traditional green roof can exert a load of up to 250 kg per m$^2$; however, the widely used sedum-based ones are 6 times lighter on an average and are suitable for most buildings; specific guidelines can be found in [5,6]), which is likely to encourage their massive diffusion. This transformation of the roofs with soil and vegetation is widely believed among the research community to contribute to counter the urban environmental problems mentioned above [7].

After major technological developments, various scientific communities (e.g., hydrology, climate, ecology) have quite recently taken an interest in this field of study in order to evaluate its performances, in particular by characterizing some of them that are sometimes approached as ecosystem services, such as the regulation of heat and water flows or urban cooling. By studying green roofs in the UK, [8] demonstrated their contribution to the insulation of buildings, with significant influence of the substrate properties but also of the actual thermal insulation systems. Refs. [9,10] showed a moderate contribution to the reduction in annual energy demands as a function of the seasons. Oppositely, under tropical climate, [11] found insignificant contribution of green roofs to insulation and energy savings. In Sweden, [12] demonstrated a positive impact of green roofs on the reduction of runoff water as a function of the depth of substrate and its moisture prior to the rain event but failed to highlight a correlation with the slope. Under various conditions, [13,14] measured a large range of water retention performance as a function of the substrates properties but also of the season and meteorological conditions. This last aspect and the consequent substrate's moisture were also strongly highlighted by [15] in three distinct climate regions. The capacity of green roofs to delay rainwater runoff has been less studied, but evidences of the potential contribution of green roofs has been noticed [16,17].

Even if authors such as [18,19] have studied the temporal evolution of green roofs, it has to be noted that most other research have been conducted considering that green roofs are inert or abiotic systems and that their properties remain constant over time. However, not only the very nature of the components of their substrates but also the functions provided by green roofs require to consider them as soils and more specifically Isolatic Technosols, which makes them highly reactive and submitted to an early pedogenesis [20]. As a result, a significant evolution of the porous architecture of substrates over time is noted, modifying their internal properties [21]. Beyond that, additional results suggest that these changes in such properties lead to variations in the performance levels of green roofs [16,22].

Our paper aims to describe the influence of factors and properties of green roofs by extracting such factor–property–performance relations (defined in Section 2.1) from the previously conducted studies. Indeed, some researchers conducted various in-situ or controlled-conditions experiments or modelling approaches in order to study performances—at a certain time, some on ageing, some on the influence of the properties of materials. With such a diverse literature, meta-analysis can serve as a useful tool to parse through both main findings until now to moderate and evaluate what works best for a particular scenario and the ways to improve that approach for future studies. It can also serve as a go-to bibliography for further relevant studies. A fruitful review has already been conducted that showed the wide diversity of performances of green roofs across the world [23]. Our expected additional contribution is to qualify the factors and properties that would influence the performances of green roofs in order to: (i) highlight the most relevant levers that could be used to enhance their contribution to urban environmental issues and (ii) suggest future directions for research on green roofs.

## 2. Materials and Methods

### 2.1. Definitions and Frame of the Analysis

**Performances** of a green roof are the direct and indirect contributions and benefits to human well-being [24–26]; more specifically, it describes the ability of a green roof to provide ecosystem services [26]. Some authors also mentioned them as "operational environmental cost-benefits" [27]. In this work, we focused our attention on hydrological and thermal performances (Table 1). Such performances are supposed to be under the influence of factors and properties.

**Table 1.** Final performances considered.

| Hydrological Performances | Thermal Performances |
|:---:|:---:|
| Retention | Insulation |
| Detention | Energy demand |
| | Reduction of surface temperature |
| | Inner building temperature |

**Retention** is defined as the capacity of a green roof to limit the quantity of drained water as a function of the incoming water (rainfall) [4,20] In other terms, it is "the amount of stormwater that will not become runoff [of] the roof" [28]. The water is both retained into the substrate (due to capillary forces) or the drainage layer (depending on its configuration) and is later evaporated or uptake and transpired by vegetation. **Detention** is the capacity of a green roof to delay the peak of discharge of drained water as a function of the start of the rainfall thanks to its flow into the substrate and the drainage layer [20,24,28,29].

**Insulation** is the ability of the roof to "reduce the heat flux through a building envelope since the growing medium acts as an insulating layer" [25]. **Energy demand** of a building is the amount of energy used for cooling and heating [25]. **Reduction of surface temperature** is the ability of green roof to reduce the roof surface temperature, thereby helping in the reduction of energy needs of the building. Inner building temperature is the building's interior temperature, which can be dependent on the efficiency of green roof to prevent solar radiation from heating interior spaces of buildings [25].

From an historical soil science perspective, we chose to very clearly distinguish the characteristics from the environment of green roofs designated as factors and the characteristics of the green roof system designated as properties according to the seminal work from [30].

**Factor** is an external characteristic of the environment of the green roof that may influence and affect its performances [27]. They are related to weather conditions (e.g., air temperature, wind conditions, humidity); to the characteristics of rain events, such as its intensity, frequency, and duration; or related to building characteristics [4]. A total of 25 factors were initially considered and progressively restricted to 12 (Table 2). The restriction was done after going through the available research articles and analysis the relevancy of the factors and availability of research material for the particular factor.

**Property** is here specifically defined as an internal characteristic of the green roof that may affect its performances. The properties can be related to the number of layers; the global, physical, and chemical characteristics of the substrate (e.g., depth, composition, porosity); and of the vegetation (type, density) [4]. It has to be noted that certain authors named these properties as "structural factors" [31–33]. A total of 33 properties were initially considered and progressively restricted to 18 (Table 3), similar to the methodology considered in restricting factors.

**Table 2.** Final factors considered.

| Factors | |
|---|---|
| | Rain intensity |
| Rain characteristics | Rain frequency |
| | Rain duration |
| Atmospheric conditions | Air humidity |
| | Wind speed |
| | Average temperature |
| Temperature characteristics and heat fluxes | Maximum temperature |
| | Solar radiation |
| | Latent heat |
| | Building slope |
| Building characteristics | Building height |
| | Nature of covering material |

**Table 3.** Final properties considered.

| Properties | |
|---|---|
| Age of the green roof | Date of implementation |
| | Substrate composition |
| Global substrate characteristics | Substrate depth |
| | Horizonation/layering |
| | Porosity |
| | Granulometry |
| Substrate's physical characteristics | Bulk density |
| | Solid density |
| | Water holding capacity |
| | Hydraulic conductivity |
| Substrate's thermal characteristics | Thermal resistance |
| | Solar reflectivity |
| | Percentage of vegetation |
| Vegetation characteristics | Nature of vegetation |
| | Leaf area index |
| | Diversity of plant species |
| Biological activity | Soil fauna |
| | Soil micro organisms |

Tables depicting initial factors and properties considered before finalization are mentioned in Table A1 of Appendix A.

The term "**performance relation**" is used in order to express the way the studied factors and properties influence the different hydrological and thermal performances of the green roof. *FPR* refers to factor-performance relation, such as effect of rain intensity on retention. *PPR* refers to property-performance relation, such as effect of substrate depth on retention. *FPPR* refers to combined factor–property–performance relations. Each influence of a factor on a performance studied by an author is considered as one FPR; similarly, each influence of a property on a performance studied by an author is considered as one PPR.

Hence, a same study and a single experiment had the possibility to result in multiple FPR and PPR.

**Influence indices (positive-P, negative-N, neutral-O)** are the ways the factor/properties influence the performances. A positive influence would mean that the factor/property increases the performance provided by the green roof system. For example, an increase of the substrate depth was demonstrated to increase the retention capacity [31]; hence, the property "substrate depth" will be considered by these authors as positive on the performance "retention". On the contrary, an intense rainfall can reduce the retention capacity of a green roof [34,35]; hence, the factor "rainfall intensity" will be considered by these authors to have a negative influence on the performance "retention". Determination of the influence indices was done by evaluating the details of the experimental observations of all studied papers and the way the authors expressed the influence to be positive/negative/neutral on any performance. Different papers could find different influence indexes for the same FPPR depending on what aspect of the performance the studied author concentrated on.

The green roof system can therefore be seen as a complex system that could be described by its "properties", considering on the one hand the nature and physical, chemical, and thermal properties of its abiotic components (i.e., substrate and drainage layers) and on the other hand its biotic components (i.e., planted or seeded and spontaneous vegetation, spontaneous fauna, and microbiota) (Figure 1). As in all biotic/abiotic systems, complex interactions happen. First, the external system—here described as "factors"—induces an ageing effect that results in the evolution of "properties" over time (e.g., rain may induce leaching of fine particles; cold temperature may alter the vegetation development). Moreover, interactions between abiotic and biotic components may also induce evolution of "properties" (e.g., plant litter may increase the organic matter content in the substrate; decrease of the substrate physico-chemical fertility can decrease the biomass production).

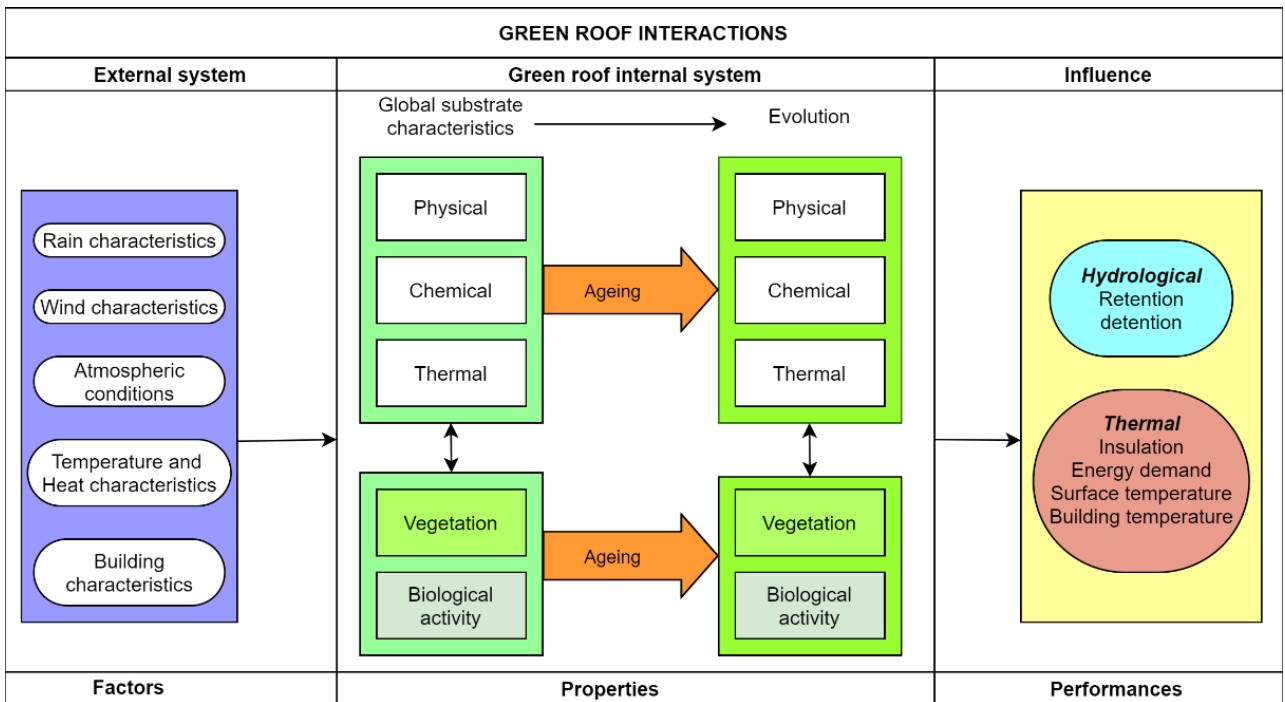

**Figure 1.** Green roof system's interactions between its global characteristics, internal properties, and external factors.

Eventually, it can be said that such inter-relations and interactions between all "factors" and "properties" control the level of performances that could be submitted to changes over time. Though the system is complex, our approach sheds light on the potential of simplifying each FPPR and hence understanding the system evolution and performances better (Figure 1).

## 2.2. Search Strategy

To find the articles that studied these FPPR, the Web of Science database was the chosen source of the data from the three initially considered sources (Web of Science, ScienceDirect, and Google scholar). The keywords were selected based on their popularity of utilisation among the authors in this field by trying out various possible and relevant words related to hydrological and thermal performances. The search was conducted using the following terms: [TS = (green roof* AND hydrological performance), TS = (green roof* AND hydric performance), TS = (green roof* AND thermal performance), TS = (green roof* AND thermic performance)].

The reference list of identified papers was initially manually checked for their relevancy based on their title and keywords and were filtered suiting the purpose of this paper (hydrological and thermal performances). After the relevant articles were selected (number of selected papers, N = 395), their abstracts and type of study were analysed, and any review/non-experimental based articles were eliminated (N = 187).

The remaining studies (N = 208) were selected if they met the following criteria:

- The study conducted has at least one experiment/modelling approach where the authors studied at least one factor/property and tested the effect of this factor/property by comparing different values of it and directly mentioning the influence of factor/property on the hydrological/thermal performance of green roof and their influence index.
- Studies were excluded if they lacked any relevant information, either missing or ambiguous, as per the understanding of the author. Any other publication lacking primary data and/or explicit method descriptions were also excluded.

As a result, 100 articles were considered for the rest of this work. These are mentioned in Table A2 of Appendix B.

## 2.3. Data Extraction

Data extracted from the eligible studies were the FPR and PPR and their influence indices. Other than this, the following information was also noted: (i) first author's name, (ii) first author's expertise (based on the theme of the published journal), (iii) journal aims and scope, (iv) year of publication, (v) study location, (vi) study design, (vii) climatic condition, (viii) type of study, and (ix) keywords.

## 3. Results

### 3.1. Characteristics of the Included Studies

Eligible studies were published between 1997 and 2020, with the biggest chunk of the studies in the past decade (80%) (Figure 2).

Majority of the studies were sourced from four countries, including the USA (12); the UK (11); France (9), whose research has been consistent throughout the past decade; and China (11), which has had a boom in green roof research in the last three years. Other than these, to a lesser extent, studies have also been conducted in Italy, Sweden, Spain, Belgium, the Netherlands, Madagascar, Norway, Brazil, Turkey, and Canada during the past decade.

While countries from Europe and North America have been contributing steadily throughout the past two decades, countries mainly from sub-tropical climate regions and Asian origin, such as China, Malaysia, Japan, Australia, Singapore, Israel, and Iran, have contributed significantly during the last half of the decade, thereby almost tripling the overall research conducted in this area. Before this sub-tropical influx, most of the studies were from the temperate maritime and continental regions.

Almost half of the studies (47%) were conducted using modelling approach (mostly since 2017), whereas 34% were in-situ experiments, and 19% were lab experiments (Figure 2). Until 2011, there was frequent but little research done as per our relevance. Since 2011, there has been a gradual increase, especially in modelling studies.

Though it is difficult to pinpoint the first author's expertise of the articles, it can be said that most of them are of hydrology background, followed by energy engineering, environment, soil, and horticulture, among other similar fields.

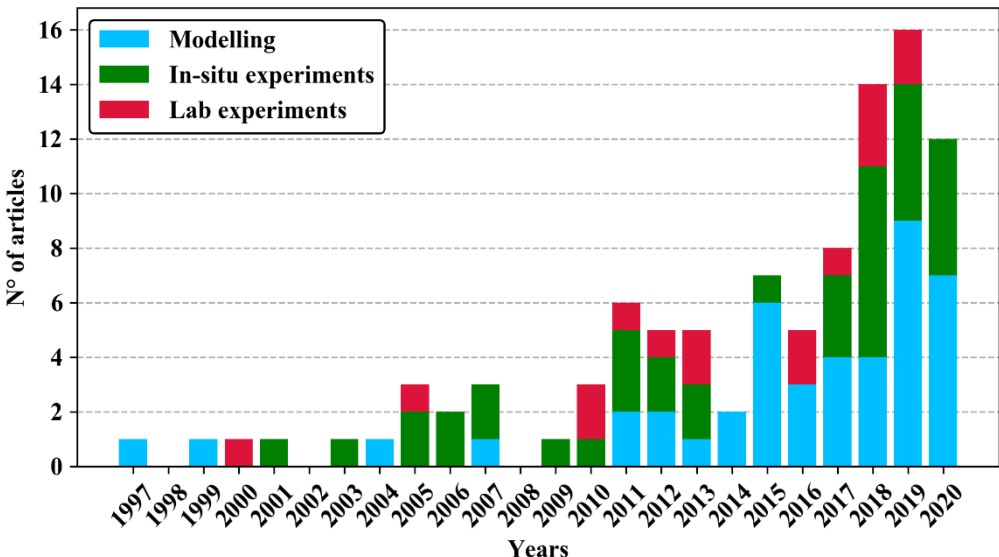

**Figure 2.** Yearly evolution of type of studies (modelling, in-situ experiments, and lab experiments) conducted.

### 3.2. Total Number of External Factors and Internal Properties (Considered vs. Studied)

Out of the 25 factors initially considered, based on the criteria explained in the materials and methods section, 12 factors were studied at least by a single article. Similarly, compared to external factors, more internal properties (33) were initially considered, and more (18) properties were actually studied by the authors. Properties related to the set-up of the green roof system (e.g., composition, substrate depth) and vegetation were comparatively well studied, whereas chemical properties, such as nitrogen content or chemical conductivity, were ignored by almost all the authors considered.

### 3.3. Influence of External Factors

Empty spaces are prominent (70.8%) in the FPR heat map (Figure 3), showing limited cases have been considered by the articles and that not all FPR are studied. In fact, overall studies conducted are also less than half that of the PPR. The same percentage of FPR (29.2%) for both the hydrological performances (7 out of 24 FPR studied) and thermal performances (14 out of 48) are studied. However, among the studied ones, thermal performances are studied better, with an average of three studies per FPR. Reduction of surface temperature is the best-studied thermal performance, with 15 FPR studies. Among the hydrological performances, retention is well studied, with 19 FPR. Detention and insulation performances are almost ignored. It can also be noted that there is not even a single specific factor that has been studied well for both hydrological and thermal performances. Factors related to rain have been decently studied for retention performance, with rain intensity's effect on retention being the most studied FPR, with seven studies. The studied factors are also considered to a much lesser extent, with only six FPR studied more than four times overall. On an average, each nonempty FPR is studied by 2.4 publications.

Only two factors (solar radiation and height of the building) have been studied for their influence on at least three performances. The average here is two.

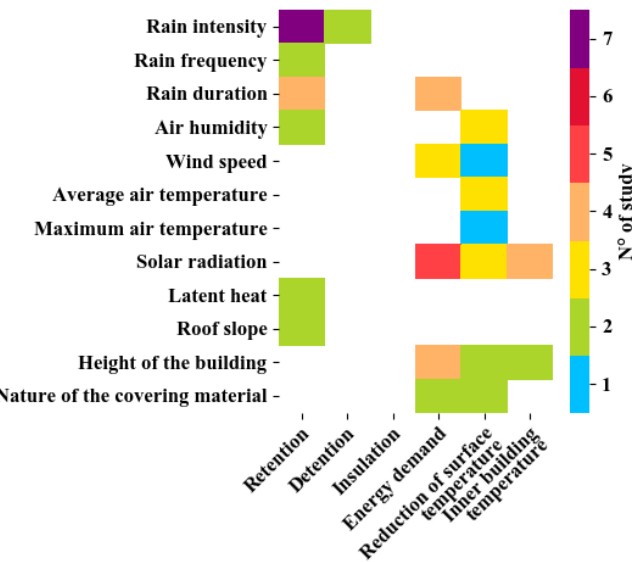

**Figure 3.** Heatmap depicting external factors influence on performances (FPR).

*3.4. Influence of Internal Properties*

Even if empty cells are still found in the PPR Heatmap (Figure 4), they are much less prominent compared to FPR (42%), suggesting that they have been better studied.

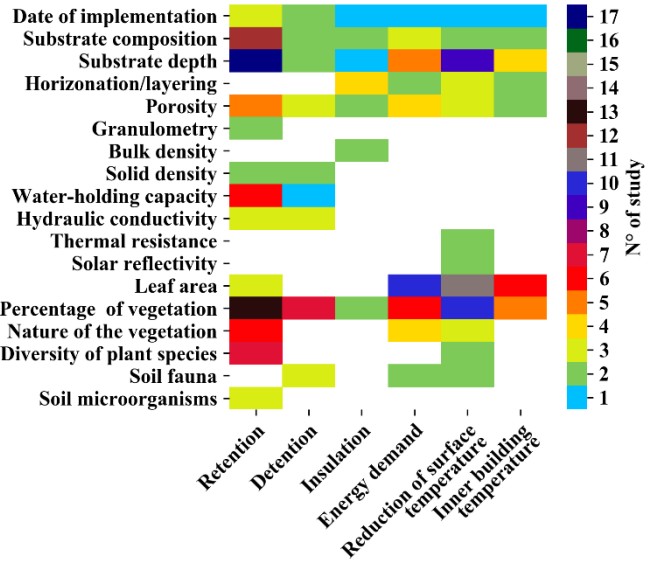

**Figure 4.** Heatmap depicting internal properties influence on performances (PPR).

The performances are more uniformly studied, with at least five properties being studied for all the performances. Among the hydrological performances, retention and detention are equally well studied. Thermal performances have been less studied, with the reduction of surface temperature being the best studied. Relating to substrates, substrate nature and physical properties, in particular composition and depth, have been studied the most by the authors overall considering all the performances, with it being mentioned 26 times and 38 times, respectively. On an individual scale, substrate depth's influence on the retention performance is the most studied PPR, with authors studying it 17 times.

Relating to vegetation, two properties (percentage of vegetation and leaf area index) have been studied for their influence on at least four performances. The average here is two. Among the ones studied, percentage of vegetation has been studied the most by the authors overall with respect to performances, with it being studied for 40 PPR, making it

the most studied property. On an individual relation scale, leaf area index's influence on the reduction of surface temperature and percentage of vegetation's influence on retention are the most studied PPR, with authors studying it 11 times and 13 times, respectively. In a rather contrasting manner, while substrate and vegetation properties are vastly studied for all the performances, others have had a great decline, with them being studied for just one or two performances. On an average, each nonempty PPR is studied by 2–3 authors.

By considering six performances and a final total of 30 factors and properties, there was a possibility of 180 FPPR studies overall. As presented previously, out of these possibilities, only 82 have been studied at least once, leaving the other 98 relations (54.4%) unexplored.

### 3.5. Factor–Performance Relations Influence

It can be seen that the studied factors have a major negative influence on the performances, especially the hydrological performances, owing to the rainfall prominently (e.g., the highest is the rainfall intensity; the worst is the retention performance of the green roof) (Figure 5). In fact, 90% of the hydrological FPR have negative influence. With regard to thermal influence, the positive influences are more noticeable, with 45% and 55% being negative. Only one FPR was found to have a neutral influence. Building-temperature-related FPR has the most positive influence, with 55%.

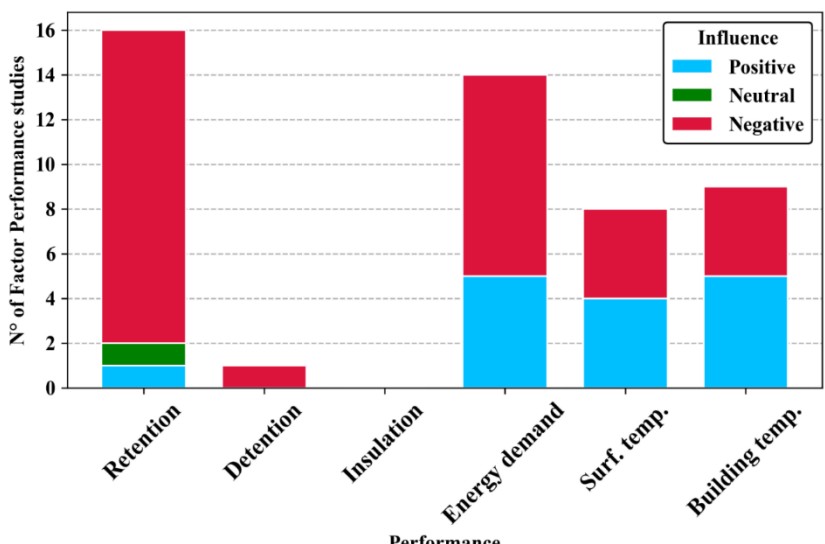

**Figure 5.** Influence index of FPR on respective hydrological and thermal performances, showing largely negative influence.

In general, it can be seen that most studies are in accordance with each other on the type of influence. In one case (building slope), there is a difference in opinion among the studies with respect to their influence in the hydrological-retention performance study.

### 3.6. Property–Performance–Relations Influence

Contrary to the influence of factors, the studied properties have an almost completely positive influence (95%). Results are very consistent for all performances regardless of whether they are hydrological or thermal or regardless of whether they are very well studied (such as retention) or less (such as insulation) (Figure 6).

The difference between various authors in type of influence indices in the PPR studies is higher than that of FPR, with 5 out of 18 PPR having differences in their type of influence. Here, along with four cases of retention (diversity, percentage of vegetation, composition, and water-holding capacity), there is also one case of insulation (layering) where difference arises.



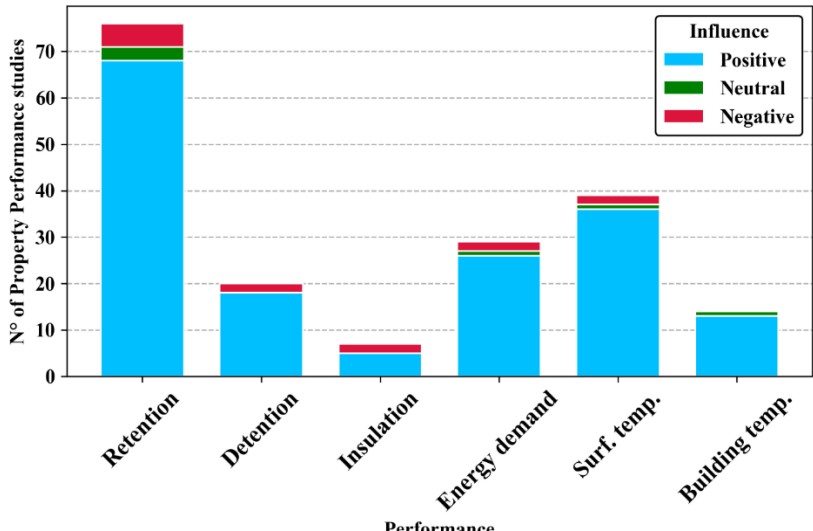

**Figure 6.** Influence index of PPR on respective hydrological and thermal performances, showing largely positive influence.

Percentage of vegetation, the highest studied PPR, has only one disagreement (in retention). Leaf area index, porosity, and substrate depth, the other well-studied PPR, have 100 percent homogenous observations, each being studied for all the performances and having positive influence on all of them. Diversity of vegetal species is the most diverse FPR with respect to influences, with studies showing it can influence retention performance positively, negatively, and neutrally, too.

Overall, with only five FPR studies among all the studies not being in accordance with others, it can be said that the type of influence is highly homogenous.

## 4. Discussion

### 4.1. Identification of Levers to Enhance the Performances of Green Roof

Our main findings highlighted that most properties have a positive influence on the performances of green roofs, showing there are many existing levers to enhance green roof performances and tackle some of the main issues already mentioned, i.e., flooding and urban heat island.

A few of these could easily be controlled by manufacturers and have been largely mentioned already, such as increasing the depth of the substrate, which could help increasing the retention and detention of water alongside contributing to insulation and consequently to the decrease of energy demand and inner building temperature in hot periods. Substrate composition and layering of various materials are undoubtedly another key parameter to control the level of performances provided by green roofs [36,37]. However, the relations between them and performances remain complex to establish, considering that it is highly dependent on many internal properties (nature and proportion of the substrate's parent materials) [38] and external factors, such as the climatic conditions [29,39]. Further research that could help in predicting the contribution of various components and their association on green roof performances would be highly needed. Vegetation also appears as a valuable lever to increase performances in relation with their physiology (i.e., transpiration, shadowing, biodiversity habitat). Authors such as [40] have already mentioned the importance of choosing local and native plants and of maintaining an optimal vegetation development through a possible input of fertilizers [23].

Some other properties have a significant impact on performances but are more difficult to handle, as they are not directly manageable. For example, the higher the porosity, the higher the retention and detention of water [35,41] and the better the insulation [42]. Similar thoughts could be made on bulk density or water-holding capacity. However, such physical characteristics rely not only on the properties of each constituent of the

substrate but also on the result of their formulation to design the substrates. In addition, a change over time in the chemical (lower pH, higher organic carbon, and total nitrogen levels) and physical (settlement) characteristics of different green roof substrates was identified by [43] without the consequences of these changes on hydraulic properties being measured. As a consequence, further research on the physical, hydrological, and thermal properties of substrate would be required in order to enhance the way they control the behaviour of green roofs [31,44]. More globally, the relations between substrates fertility (i.e., concentration in nutrients, organic matter content, water available for plants) are also strongly correlated with the all characteristics related to vegetation. As a consequence, an agronomical approach of green roofs appears promising to describe the interrelations of such properties.

Finally, factors are, by definition, absolutely unmanageable, as they are dependent on either climatic, meteorological, and building properties or the ecological continuum [29,45]. Based on that, a deep work on the adaptation of the properties of a green roof to its environment should be promoted [46].

### 4.2. Filling the Gaps: The Exploration of the Unstudied Relations

Previous experimental and modelling research has explored various relations between factors or properties and hydrological or thermal performances of green roofs. However, more than half of the potential relations we identified—considering that our lists were far from exhaustive—remains unexplored. Even if they do not have all the same relevancy, many outlooks emerged from such a review analysis. Ref. [47], in their study, highlighted the need and complexity of green roof designing due to unmanageable factors and time- and location-specific constraints. As more and more FPPR are explored, the complexity of green roof becomes more manageable with regards to designing for optimal performance. This is mainly possible, as through this approach instead of the whole system redesign, specific factors and properties can be controlled and monitored. Most research through the years, such as [39,48], have also highlighted the need for specific approaches and expansion of the green roof horizon. But it seems that due to design and implementation constraints, only the tried and tested approaches, factors, and properties are preferred.

Concerning the impacts of the studied factors, it appears that the ones that are related to hydrology (i.e., rain characteristics) are dominantly separated from the ones related to heat fluxes. Indeed, the combined effects of thermal and hydrological behaviour of the green roof on its performances remain largely unknown. For example, the effect of dry conditions prior to a rain event are known to have an impact on both water and heat fluxes into the substrate as well as, more specifically, the impacts of the temperature or the wind speed on the plant transpiration [39]. At the scale of the whole building, the consequences of its various characteristics are complex to capture with experiments and mainly rely on modelling approaches, which require specific skills [49].

The study of green roof properties appears complex, as all the mentioned characteristics are not only correlated but also evolving over time. The very nature of artefacts, dominant components of green roof substrates, are well known to be extremely reactive to environments, which can result in intense and rapid changes in their properties [50]. Apart from its own effects, vegetation could lead to an evolution of Technosols' physicochemical characteristics [51], and so could the soil fauna and even microorganisms. Various studies, such as [16,22], already suggested that the temporal changes in properties lead to variations in the hydraulic performance levels of green roofs. Again, relations between various chemical properties of green roof substrates (e.g., relative abundancy of organic and mineral contents, availability of nutrients) and their dynamic over time that control the development of vegetation appear as a major gap in the study of the green roof performances. As an explicit example, fine particles' eluviation from substrates surface have been monitored on green roofs, which may lead to physical constraints for plant growth [20].

There is a need for studies that consider both approaches and reach a crossroad point between ageing—which is the term dedicated to evolution of inert materials—and

pedogenesis—which is the term that describes evolution of living media—to know the similarities, their interaction, as well as inter-relation between these two approaches.

*4.3. Promotion of Deeper and Innovative Experimental Approaches for Research on Green Roof Performances*

Everybody seems to agree and appreciate that green roofs are beneficial, but as seen in this article, scientifically speaking, there is very less validation with regard to it yet. There are many studies about green roofs that were not adequately designed on purpose to study the dedicated effect of factors or properties, owing to the fact that they either considered the green roof as a whole system, such as [52], or limitations in specific approach development, such as [53]. There is a need to develop specific approaches relevant and customized to the green roof scale and behaviour to explore the green roof potential to the fullest extent possible. In their review, [4] clearly echoed the same sentiment: that there is a need for more research into green roof performance in an urban environment, as the differences measured by few existing studies between the early years' performance of green roofs and the later years indicate a need for long-term monitoring of green roofs. Ref. [28] also highlighted there is scarcely any literature on the operation of full-scale, building-implemented green roofs, and no articles were found on the building technical performance of aged green roofs. That is not only a risk factor that fails to evaluate long-term green roof performance but also an utter disregard to the potential improvements in efficiency of green roof performances by not exploiting it.

In-situ experiments are very rich in terms of data that could be monitored and harvested under real conditions [54,55]. Therefore, it is consistently prominent in the past decade considering its longevity. On the downside, they need long-term observation and adapted monitoring devices to be implemented at the beginning, and hence, even if large-scale data are available, their exploitation can be time consuming, tedious, and impractical. Therefore, the definition of some shared protocols and knowledge about good practices is expected. Then, implementing long-term observation sites under various situations would also serve as a reference to all the concerned studies. However, it is found that the vast majority of in-situ green roof research conducted has been on smaller green roof test beds or isolated components [28]. This is an interesting observation because, as this paper highlights, pedogenesis and evolution of green roof components have a great potential to modify the performance over time. However, the test beds are usually discarded/not maintained after the experiments, hence denying the possibility of temporal evolution analysis.

Comparatively, lab-scale experiments are promising [16,17,20] as they, under controlled conditions, could reproduce in a very specific way some variations of factors and properties. However, they might not be multipurpose and very specific to certain properties or factors. Unfortunately, this is the least-utilized method for now considering the difficulties in designing them. Hence, there is a need to design and create such lab prototypes to understand the influence of each relation and also in order to lighten the long-term experiments, too.

Considering various implementation and monitoring complexities in the other two approaches, the modelling approach has been the go-to suggestion by most researchers and reviewers in the past three decades, showing its potential [39,48]. However, the existing modelling approaches are not perfectly suited to the evaluation of green roof performances, as they are more focused on the thermal and hydrological processes (i.e., transfer and fluxes). Ref. [56] noted that currently, there are many mathematical models in green roofs, but as each one of them has their own notation, it complicates the study of this field. Even if modelling approaches have the liberty to simplify systems, a more realistic and green-roof-specific consideration can be adapted. For example, modelling approaches do usually consider 100 percent vegetation on green roofs, which is rarely the case owing to maintenance issues and easy foreign matter growth. Ref. [48] also highlighted the need for a green-roof-specific approach because of the complex interactions between the hydraulic and thermal processes that occur in these systems.

*4.4. From Mono to Transdisciplinary Research about Green Roofs*

Throughout the study, it was noted that all the research was made by the authors who had specific interests and studied mostly that specific aspect. Their interest would be either about hydrological performances or thermal performances, rarely considering both performance groups together and not specifically related to green roofs [57]. A broader, transversal, and complex study could lead to a more reliable and precise estimation and assessment of FPPR.

The expertise and interests of the published authors have also confined them to more prominent properties, such as the substrate composition and the vegetation. They have been demonstrated to have direct influence on the performances, and hence, most of the studies concentrated on them, as they were easier to characterize, ignoring the other less common but possibly impactful study possibilities. Despite the study of many different substrates with various composition, almost no research has yet been conducted to establish the clear relations between physical properties (e.g., bulk density, particle size of the components) or chemical characteristics (e.g., organic matter or nitrogen contents) and performances [58,59].

Other disciplines could help in understanding the various properties and factors that influence the behaviour of green roofs, such as ecology, biology, soil science, agronomy, and architecture [60]. In this regard, [23] suggested a research on finding the coefficients of the impact of green roofs to universalize the effect of green roofs on performances, as the results presented by different authors (most often based on a single or limited number of case studies) differ significantly from each other, owing to change in the climate, geography, study concerned, etc.

## 5. Conclusions

This paper first contributed to the definition of a framework likely to be considered to describe the intrinsic complexity of green roofs as biotic–abiotic systems—at the crossroad between architecture and horticulture—and its relation with external factors, notably meteorological conditions, that contributed to multiple ecosystem services, here focused as hydrological and thermal performances. As part of the drastically increasing scientific interest for green roofs, our meta-analytic review highlighted many researches dedicated to the impact of green roof properties and factors on such performances. Even if these properties–performance relations are seemingly more important, the comparative ignorance of more than half of the other possible factors–properties relations also holds important implications considering their potential in defining green roof performances. It could also be seen that the studied factors have a major negative influence on the performances (68%), whereas the studied properties have a major positive influence (95%), which highlights that even if the green roof is a complex system, there exists different levers to be considered for performances enhancement. The trend of more and more researchers opting for modelling approaches in the past decade also paves way for further studies with promotion of deeper and innovative experimental approaches for research on green roof performances, especially in real site conditions to understand the performance changes in a realistic way. Even if this can be a tedious job considering the difficulties in design, possible inter-relations, and time consumption, it is the need of the hour considering their possible impact in improving green roof performances and easing the design process. The first step regarding this could be developing clear protocols for particular factors–properties–performance studies to simulate real site conditions. This can ideally be suitable to be able to implement in multiple climatic conditions by modifying certain parameters based on the regional characteristics. Along with this, there is a necessity to develop long-term observations for better understanding and control of the evolutionary and early paedogenic performance changes.

**Funding:** This work is financially supported by the Centre for Studies and Expertise on Risks, Mobility, Land Planning, and the Environment (Cerema).

**Institutional Review Board Statement:** Not applicable.

**Informed Consent Statement:** Not applicable.

**Data Availability Statement:** Not applicable.

**Conflicts of Interest:** The authors declare no conflict of interest.

## Appendix A

**Table A1.** Tables depicting initial factors and properties considered before finalization.

| Initial Factors Considered | |
| --- | --- |
| **Class** | **Characteristics** |
| Rain | intensity |
| | frequency |
| | duration |
| | hydrometeor |
| Temperature | air temperature |
| | average |
| | minimum |
| | maximum |
| Heat fluxes | solar radiation |
| | conductive flux from the building |
| | infrared radiation |
| | latent heat |
| | sensible flux |
| | storage flux |
| Building surface | evenness |
| | slope |
| | total surface |
| | height of the building |
| | architecture |
| | roughness (surrounding buildings) |
| | nature of the covering material |
| Wind | speed |
| | orientation |
| Atmospheric and weather conditions | air humidity |
| | fine particles deposition |
| | cloudiness |
| | pollutants concentration |
| | greenhouse gases |

**Table A1.** *Cont.*

| Initial Properties Considered | |
| --- | --- |
| **Class** | **Characteristics** |
| Nature | composition |
| | manufacturer |
| | date of fabrication |
| | date of implementation |
| Physical | depth |
| | surface state |
| | horizonation/layering |
| | porosity |
| | granulometry |
| | bulk density |
| | solid density |
| | structure |
| | aggregate stability |
| | Pore-size distribution |
| Hydric | retention curve |
| | water-holding capacity |
| | wilting point |
| | saturation point |
| | hydraulic conductivity |
| | preferential flow |
| | van Genuchten parameters |
| Chemical | organic carbon |
| | nitrogen |
| | pH |
| | CEC |
| | available phosphorus |
| | mineralogy |
| | chemical conductivity |
| Thermic | thermal conductivity |
| | thermal capacity |
| Biological | germination capacity |
| | microbial diversity |
| | microbial abundancy |
| Living organisms (vegetation, animals, microorganisms) | organic matter addition (e.g., faeces deposition) |
| | organic matter transformation |
| | burrowing activities |
| | leaf area |
| | biological diversity |
| | biological abundancy |

## Appendix B

**Table A2.** Table mentioning the final 100 filtered articles.

| | |
|---|---|
| 1. | A. Graceson, M. Hare, J. Monaghan, and N. Hall, "The water retention capabilities of growing media for green roofs," *Ecological Engineering*, vol. 61, pp. 328–334, Dec. 2013, doi:10.1016/j.ecoleng.2013.09.030. |
| 2. | W. Liu, Q. Feng, W. Chen, W. Wei, and R. C. Deo, "The influence of structural factors on stormwater runoff retention of extensive green roofs: new evidence from scale-based models and real experiments," *Journal of Hydrology*, vol. 569, pp. 230–238, Feb. 2019, doi:10.1016/j.jhydrol.2018.11.066. |
| 3. | Liu and Chui, "Evaluation of Green Roof Performance in Mitigating the Impact of Extreme Storms," *Water*, vol. 11, no. 4, p. 815, Apr. 2019, doi:10.3390/w11040815. |
| 4. | S. Poë, V. Stovin, and C. Berretta, "Parameters influencing the regeneration of a green roof's retention capacity via evapotranspiration," *Journal of Hydrology*, vol. 523, pp. 356–367, Apr. 2015, doi:10.1016/j.jhydrol.2015.02.002. |
| 5. | M. E. Sia, "Evapotranspiration from extensive green roofs: influence of climatological conditions, vegetation type, and substrate depth," p. 88. |
| 6. | R. Bouzouidja, G. Rousseau, V. Galzin, R. Claverie, D. Lacroix, and G. Séré, "Green roof ageing or Isolatic Technosol's pedogenesis?," *Journal of Soils and Sediments*, vol. 18, no. 2, pp. 418–425, Feb. 2018, doi:10.1007/s11368-016-1513-3. |
| 7. | S. De-Ville, M. Menon, X. Jia, G. Reed, and V. Stovin, "The impact of green roof ageing on substrate characteristics and hydrological performance," *Journal of Hydrology*, vol. 547, pp. 332–344, Apr. 2017, doi:10.1016/j.jhydrol.2017.02.006. |
| 8. | S. De-Ville, M. Menon, X. Jia, and V. Stovin, "A Longitudinal Microcosm Study on the Effects of Ageing on Potential Green Roof Hydrological Performance," *Water*, vol. 10, no. 6, p. 784, Jun. 2018, doi:10.3390/w10060784. |
| 9. | Z. Peng, C. Smith, and V. Stovin, "The importance of unsaturated hydraulic conductivity measurements for green roof detention modelling," *Journal of Hydrology*, vol. 590, p. 125273, Nov. 2020, doi:10.1016/j.jhydrol.2020.125273. |
| 10. | V. Stovin, S. Poë, S. De-Ville, and C. Berretta, "The influence of substrate and vegetation configuration on green roof hydrological performance," *Ecological Engineering*, vol. 85, pp. 159–172, Dec. 2015, doi:10.1016/j.ecoleng.2015.09.076. |
| 11. | M. H. N. Yio, V. Stovin, J. Werdin, and G. Vesuviano, "Experimental analysis of green roof substrate detention characteristics," *Water Science and Technology*, vol. 68, no. 7, pp. 1477–1486, Oct. 2013, doi:10.2166/wst.2013.381. |
| 12. | A. Nagase and N. Dunnett, "Amount of water runoff from different vegetation types on extensive green roofs: Effects of plant species, diversity and plant structure," *Landscape and Urban Planning*, vol. 104, no. 3–4, pp. 356–363, Mar. 2012, doi:10.1016/j.landurbplan.2011.11.001. |
| 13. | T. Susca, S. R. Gaffin, and G. R. Dell'Osso, "Positive effects of vegetation: Urban heat island and green roofs," *Environmental Pollution*, vol. 159, no. 8–9, pp. 2119–2126, Aug. 2011, doi:10.1016/j.envpol.2011.03.007. |
| 14. | A. Albatayneh, D. Alterman, A. Page, and B. Moghtaderi, "Renewable Energy Systems to Enhance Buildings Thermal Performance and Decrease Construction Costs," *Energy Procedia*, vol. 152, pp. 312–317, Oct. 2018, doi:10.1016/j.egypro.2018.09.138. |
| 15. | E. Axelrad, "in Environmental Health Sciences," p. 38, 2019. |
| 16. | N. D. VanWoert, D. B. Rowe, J. A. Andresen, C. L. Rugh, R. T. Fernandez, and L. Xiao, "Green Roof Stormwater Retention," *Journal of Environment Quality*, vol. 34, no. 3, p. 1036, 2005, doi:10.2134/jeq2004.0364. |

**Table A2.** *Cont.*

| 17. | J. Mentens, D. Raes, and M. Hermy, "Green roofs as a tool for solving the rainwater runoff problem in the urbanized 21st century?," *Landscape and Urban Planning*, vol. 77, no. 3, pp. 217–226, Aug. 2006, doi:10.1016/j.landurbplan.2005.02.010. |
|---|---|
| 18. | F. Viola, M. Hellies, and R. Deidda, "Retention performance of green roofs in representative climates worldwide," *Journal of Hydrology*, vol. 553, pp. 763–772, Oct. 2017, doi:10.1016/j.jhydrol.2017.08.033. |
| 19. | T. Savi *et al.*, "Does shallow substrate improve water status of plants growing on green roofs? Testing the paradox in two sub-Mediterranean shrubs," *Ecological Engineering*, vol. 84, pp. 292–300, Nov. 2015, doi:10.1016/j.ecoleng.2015.09.036. |
| 20. | C. F. Fang, "Rainwater retention capacity of green roofs in subtropical monsoonal climatic regions: a case study of Taiwan," Pisa, Italy, Jun. 2010, pp. 239–249, doi:10.2495/DN100211. |
| 21. | M. A. Monterusso, D. B. Rowe, C. L. Rugh, and D. K. Russell, "RUNOFF WATER QUANTITY AND QUALITY FROM GREEN ROOF SYSTEMS," *Acta Horticulturae*, no. 639, pp. 369–376, Jun. 2004, doi:10.17660/ActaHortic.2004.639.49. |
| 22. | F. L. Duley and C. E. Domingo, "Effect of Grass on Intake of Water," p. 19, 1949. |
| 23. | T. L. Carter and T. C. Rasmussen, "HYDROLOGIC BEHAVIOR OF VEGETATED ROOFS," *JOURNAL OF THE AMERICAN WATER RESOURCES ASSOCIATION*, p. 14. |
| 24. | I. Schultz, D. J. Sailor, and O. Starry, "Effects of substrate depth and precipitation characteristics on stormwater retention by two green roofs in Portland OR," *Journal of Hydrology: Regional Studies*, vol. 18, pp. 110–118, Aug. 2018, doi:10.1016/j.ejrh.2018.06.008. |
| 25. | S. Li, H. Qin, Y. Peng, and S. T. Khu, "Modelling the combined effects of runoff reduction and increase in evapotranspiration for green roofs with a storage layer," *Ecological Engineering*, vol. 127, pp. 302–311, Feb. 2019, doi:10.1016/j.ecoleng.2018.12.003. |
| 26. | E. Fassman and R. Simcock, "Moisture Measurements as Performance Criteria for Extensive Living Roof Substrates," *Journal of Environmental Engineering*, vol. 138, no. 8, pp. 841–851, Aug. 2012, doi:10.1061/(ASCE)EE.1943-7870.0000532. |
| 27. | H. J. Ladani, J.-R. Park, Y.-S. Jang, and H.-S. Shin, "Hydrological Performance Assessment for Green Roof with Various Substrate Depths and Compositions," *KSCE Journal of Civil Engineering*, Feb. 2019, doi:10.1007/s12205-019-0270-4. |
| 28. | Y. He, H. Yu, P. Chen, and M. Zhao, "Thermal performance evaluation of a new type of green roof system," *Energy Procedia*, vol. 152, pp. 384–389, Oct. 2018, doi:10.1016/j.egypro.2018.09.161. |
| 29. | M. Dominique, R. H. Tiana, R. T. Fanomezana, and A. A. Ludovic, "Thermal Behavior of Green Roof in Reunion Island: Contribution Towards a Net Zero Building," *Energy Procedia*, vol. 57, pp. 1908–1921, 2014, doi:10.1016/j.egypro.2014.10.055. |
| 30. | G. Kokogiannakis, A. Tietje, and J. Darkwa, "The role of Green Roofs on Reducing Heating and Cooling Loads: A Database across Chinese Climates," *Procedia Environmental Sciences*, vol. 11, pp. 604–610, 2011, doi:10.1016/j.proenv.2011.12.094. |
| 31. | G. Heidarinejad and A. Esmaili, "Numerical simulation of the dual effect of green roof thermal performance," *Energy Conversion and Management*, vol. 106, pp. 1418–1425, Dec. 2015, doi:10.1016/j.enconman.2015.10.020. |
| 32. | K. L. Getter, D. B. Rowe, and J. A. Andresen, "Quantifying the effect of slope on extensive green roof stormwater retention," *Ecological Engineering*, vol. 31, no. 4, pp. 225–231, Dec. 2007, doi:10.1016/j.ecoleng.2007.06.004. |

**Table A2.** *Cont.*

| | |
|---|---|
| 33. | K. L. Getter, D. B. Rowe, G. P. Robertson, B. M. Cregg, and J. A. Andresen, "Carbon Sequestration Potential of Extensive Green Roofs," *Environmental Science & Technology*, vol. 43, no. 19, pp. 7564–7570, Oct. 2009, doi:10.1021/es901539x. |
| 34. | K. L. Getter, D. B. Rowe, J. A. Andresen, and I. S. Wichman, "Seasonal heat flux properties of an extensive green roof in a Midwestern U.S. climate," *Energy and Buildings*, vol. 43, no. 12, pp. 3548–3557, Dec. 2011, doi:10.1016/j.enbuild.2011.09.018. |
| 35. | E. P. DelBarrio, "Analysis of the green roofs cooling potential in buildings," p. 15. |
| 36. | I. Jaffal, S.-E. Ouldboukhitine, and R. Belarbi, "A comprehensive study of the impact of green roofs on building energy performance," *Renewable Energy*, vol. 43, pp. 157–164, Jul. 2012, doi:10.1016/j.renene.2011.12.004. |
| 37. | M. Tang and X. Zheng, "Experimental study of the thermal performance of an extensive green roof on sunny summer days," *Applied Energy*, vol. 242, pp. 1010–1021, May 2019, doi:10.1016/j.apenergy.2019.03.153. |
| 38. | S. Onmura, M. Matsumoto, and S. Hokoi, "Study on evaporative cooling effect of roof lawn gardens," *Energy and Buildings*, vol. 33, no. 7, pp. 653–666, Sep. 2001, doi:10.1016/S0378-7788(00)00134-1. |
| 39. | A. Khabaz, "Construction and design requirements of green buildings' roofs in Saudi Arabia depending on thermal conductivity principle," *Construction and Building Materials*, vol. 186, pp. 1119–1131, Oct. 2018, doi:10.1016/j.conbuildmat.2018.07.234. |
| 40. | A. O. Eriksson, "Water Runoff Properties for Expanded Clay LWA in Green Roofs," p. 98. |
| 41. | L. S. H. Lee and C. Y. Jim, "Thermal-cooling performance of subtropical green roof with deep substrate and woodland vegetation," *Ecological Engineering*, vol. 119, pp. 8–18, Aug. 2018, doi:10.1016/j.ecoleng.2018.05.014. |
| 42. | H. F. Castleton, V. Stovin, S. B. M. Beck, and J. B. Davison, "Green roofs; building energy savings and the potential for retrofit," *Energy and Buildings*, vol. 42, no. 10, pp. 1582–1591, Oct. 2010, doi:10.1016/j.enbuild.2010.05.004. |
| 43. | D. Armson, P. Stringer, and A. R. Ennos, "The effect of street trees and amenity grass on urban surface water runoff in Manchester, UK," *Urban Forestry & Urban Greening*, vol. 12, no. 3, pp. 282–286, Jan. 2013, doi:10.1016/j.ufug.2013.04.001. |
| 44. | Z. Zhang, C. Szota, T. D. Fletcher, N. S. G. Williams, and C. Farrell, "Green roof storage capacity can be more important than evapotranspiration for retention performance," *Journal of Environmental Management*, vol. 232, pp. 404–412, Feb. 2019, doi:10.1016/j.jenvman.2018.11.070. |
| 45. | A. Talebi, S. Bagg, B. E. Sleep, and D. M. O'Carroll, "Water retention performance of green roof technology: A comparison of canadian climates," *Ecological Engineering*, vol. 126, pp. 1–15, Jan. 2019, doi:10.1016/j.ecoleng.2018.10.006. |
| 46. | B. Johannessen, T. Muthanna, and B. Braskerud, "Detention and Retention Behavior of Four Extensive Green Roofs in Three Nordic Climate Zones," *Water*, vol. 10, no. 6, p. 671, May 2018, doi:10.3390/w10060671. |
| 47. | X. Wang, Y. Tian, and X. Zhao, "The influence of dual-substrate-layer extensive green roofs on rainwater runoff quantity and quality," *Science of The Total Environment*, vol. 592, pp. 465–476, Aug. 2017, doi:10.1016/j.scitotenv.2017.03.124. |
| 48. | C. Michels, S. Güths, D. L. Marinoski, and R. Lamberts, "Development of an experimental test rig for the evaluation of the thermal performance of building roofs," *Energy and Buildings*, vol. 180, pp. 32–41, Dec. 2018, doi:10.1016/j.enbuild.2018.09.023. |

**Table A2.** *Cont.*

| | |
|---|---|
| 49. | E. E. Koks, H. de, and E. Koome, "Comparing Extreme Rainfall and Large-Scale Flooding Induced Inundation Risk—Evidence from a Dutch Case-Study," in *Studies on Water Management Issues*, M. Kumarasamy, Ed. InTech, 2012, doi:10.5772/30378. |
| 50. | P. Y. Groisman, R. W. Knight, D. R. Easterling, T. R. Karl, G. C. Hegerl, and V. N. Razuvaev, "Trends in Intense Precipitation in the Climate Record," *Journal of Climate*, vol. 18, no. 9, pp. 1326–1350, May 2005, doi:10.1175/JCLI3339.1. |
| 51. | Y. Gong *et al.*, "Factors affecting the ability of extensive green roofs to reduce nutrient pollutants in rainfall runoff," *Science of The Total Environment*, vol. 732, p. 139248, Aug. 2020, doi:10.1016/j.scitotenv.2020.139248. |
| 52. | M. Eksi and D. B. Rowe, "EFFECT OF SUBSTRATE DEPTH AND TYPE ON PLANT GROWTH FOR EXTENSIVE GREEN ROOFS IN A MEDITERRANEAN CLIMATE," *Journal of Green Building*, vol. 14, no. 2, pp. 29–44, Mar. 2019, doi:10.3992/1943-4618.14.2.29. |
| 53. | Y. Dusza, "Toitures végétalisées et services écosystémiques: favoriser la multifonctionnalité via les interactions sols-plantes et la diversité végétale," p. 209. |
| 54. | Z. Zeng *et al.*, "Climate mitigation from vegetation biophysical feedbacks during the past three decades," *Nature Climate Change*, vol. 7, no. 6, pp. 432–436, Jun. 2017, doi:10.1038/nclimate3299. |
| 55. | A. K. K. Lau, E. Salleh, C. H. Lim, and M. Y. Sulaiman, "Potential of shading devices and glazing configurations on cooling energy savings for high-rise office buildings in hot-humid climates: The case of Malaysia," *International Journal of Sustainable Built Environment*, vol. 5, no. 2, pp. 387–399, Dec. 2016, doi:10.1016/j.ijsbe.2016.04.004. |
| 56. | Q. Meng, Y. Zhang, and L. Zhang, "Measurement of the Equivalent Thermal Resistance of Rooftop Lawns in a Hot-Climate Wind Tunnel," p. 6, 2006. |
| 57. | V. Sandoval *et al.*, "Impact of the Properties of a Green Roof Substrate on its Hydraulic and Thermal Behavior," *Energy Procedia*, vol. 78, pp. 1177–1182, Nov. 2015, doi:10.1016/j.egypro.2015.11.097. |
| 58. | P. Du, S. K. Arndt, and C. Farrell, "Is plant survival on green roofs related to their drought response, water use or climate of origin?," *Science of The Total Environment*, vol. 667, pp. 25–32, Jun. 2019, doi:10.1016/j.scitotenv.2019.02.349. |
| 59. | M. Zhao, P. C. Tabares-Velasco, J. Srebric, S. Komarneni, and R. Berghage, "Effects of plant and substrate selection on thermal performance of green roofs during the summer," *Building and Environment*, vol. 78, pp. 199–211, Aug. 2014, doi:10.1016/j.buildenv.2014.02.011. |
| 60. | J. Cao, S. Hu, Q. Dong, L. Liu, and Z. Wang, "Green roof cooling contributed by plant species with different photosynthetic strategies," *Energy and Buildings*, vol. 195, pp. 45–50, Jul. 2019, doi:10.1016/j.enbuild.2019.04.046. |
| 61. | H. Rakotondramiarana, T. Ranaivoarisoa, and D. Morau, "Dynamic Simulation of the Green Roofs Impact on Building Energy Performance, Case Study of Antananarivo, Madagascar," *Buildings*, vol. 5, no. 2, pp. 497–520, May 2015, doi:10.3390/buildings5020497. |
| 62. | N. H. Wong, Y. Chen, C. L. Ong, and A. Sia, "Investigation of thermal benefÿts of rooftop garden in the tropical environment," *Building and Environment*, p. 10, 2003. |
| 63. | S. Nadia, S. Noureddine, N. Hichem, and D. Djamila, "Experimental Study of Thermal Performance and the Contribution of Plant-Covered Walls to the Thermal Behavior of Building," *Energy Procedia*, vol. 36, pp. 995–1001, 2013, doi:10.1016/j.egypro.2013.07.113. |
| 64. | M. Porcaro, M. Ruiz de Adana, F. Comino, A. Peña, E. Martín-Consuegra, and T. Vanwalleghem, "Long term experimental analysis of thermal performance of extensive green roofs with different substrates in Mediterranean climate," *Energy and Buildings*, vol. 197, pp. 18–33, Aug. 2019, doi:10.1016/j.enbuild.2019.05.041. |

**Table A2.** *Cont.*

| | |
|---|---|
| 65. | S. Parizotto and R. Lamberts, "Investigation of green roof thermal performance in temperate climate: A case study of an experimental building in Florianópolis city, Southern Brazil," *Energy and Buildings*, vol. 43, no. 7, pp. 1712–1722, Jul. 2011, doi:10.1016/j.enbuild.2011.03.014. |
| 66. | A. Fitchett, P. Govender, and P. Vallabh, "An exploration of green roofs for indoor and exterior temperature regulation in the South African interior," *Environment, Development and Sustainability*, vol. 22, no. 6, pp. 5025–5044, Aug. 2020, doi:10.1007/s10668-019-00413-5. |
| 67. | G. Pęczkowski, T. Kowalczyk, K. Szawernoga, W. Orzepowski, R. Żmuda, and R. Pokładek, "Hydrological Performance and Runoff Water Quality of Experimental Green Roofs," *Water*, vol. 10, no. 9, p. 1185, Sep. 2018, doi:10.3390/w10091185. |
| 68. | C. Xu, Z. Liu, G. Cai, and J. Zhan, "Experimental Study on the Retention and Interception Effect of an Extensive Green Roof (GR) with a Substrate Layer Modified with Kaolin," *Water*, vol. 12, no. 8, p. 2151, Jul. 2020, doi:10.3390/w12082151. |
| 69. | R. M. Lazzarin, F. Castellotti, and F. Busato, "Experimental measurements and numerical modelling of a green roof," *Energy and Buildings*, vol. 37, no. 12, pp. 1260–1267, Dec. 2005, doi:10.1016/j.enbuild.2005.02.001. |
| 70. | H. He and C. Y. Jim, "Simulation of thermodynamic transmission in green roof ecosystem," *Ecological Modelling*, vol. 221, no. 24, pp. 2949–2958, Dec. 2010, doi:10.1016/j.ecolmodel.2010.09.002. |
| 71. | N. Meili *et al.*, "An urban ecohydrological model to quantify the effect of vegetation on urban climate and hydrology (UT&C v1.0)," *Geoscientific Model Development*, vol. 13, no. 1, pp. 335–362, Jan. 2020, doi:10.5194/gmd-13-335-2020. |
| 72. | N. Gerzhova, J. Côté, P. Blanchet, C. Dagenais, and S. Ménard, "A Conceptual Framework for Modelling the Thermal Conductivity of Dry Green Roof Substrates," p. 27. |
| 73. | A. Vasl, H. Shalom, G. J. Kadas, and L. Blaustein, "Sedum —Annual plant interactions on green roofs: Facilitation, competition and exclusion," *Ecological Engineering*, vol. 108, pp. 318–329, Nov. 2017, doi:10.1016/j.ecoleng.2017.07.034. |
| 74. | R. Bouzouidja, G. Séré, R. Claverie, S. Ouvrard, L. Nuttens, and D. Lacroix, "Green roof aging: Quantifying the impact of substrate evolution on hydraulic performances at the lab-scale," *Journal of Hydrology*, vol. 564, pp. 416–423, Sep. 2018, doi:10.1016/j.jhydrol.2018.07.032. |
| 75. | S. E. Gill, J. F. Handley, A. R. Ennos, and S. Pauleit, "Adapting Cities for Climate Change: The Role of the Green Infrastructure," *Built Environment*, vol. 33, no. 1, pp. 115–133, Mar. 2007, doi:10.2148/benv.33.1.115. |
| 76. | G. Ercolani, E. A. Chiaradia, C. Gandolfi, F. Castelli, and D. Masseroni, "Evaluating performances of green roofs for stormwater runoff mitigation in a high flood risk urban catchment," *Journal of Hydrology*, vol. 566, pp. 830–845, Nov. 2018, doi:10.1016/j.jhydrol.2018.09.050. |
| 77. | E. L. Villarreal and L. Bengtsson, "Response of a Sedum green-roof to individual rain events," *Ecological Engineering*, vol. 25, no. 1, pp. 1–7, Jul. 2005, doi:10.1016/j.ecoleng.2004.11.008. |
| 78. | D. D. Carpenter and P. Kaluvakolanu, "Effect of Roof Surface Type on Storm-Water Runoff from Full-Scale Roofs in a Temperate Climate," *Journal of Irrigation and Drainage Engineering*, vol. 137, no. 3, pp. 161–169, Mar. 2011, doi:10.1061/(ASCE)IR.1943-4774.0000185. |
| 79. | B. Y. Schindler, L. Blaustein, A. Vasl, G. J. Kadas, and M. Seifan, "Cooling effect of Sedum sediforme and annual plants on green roofs in a Mediterranean climate," *Urban Forestry & Urban Greening*, vol. 38, pp. 392–396, Feb. 2019, doi:10.1016/j.ufug.2019.01.020. |

**Table A2.** *Cont.*

| | |
|---|---|
| 80. | C. Y. Jim, "Effect of vegetation biomass structure on thermal performance of tropical green roof," *Landscape and Ecological Engineering*, vol. 8, no. 2, pp. 173–187, Jul. 2012, doi:10.1007/s11355-011-0161-4. |
| 81. | M. Mobilia and A. Longobardi, "Impact of rainfall properties on the performance of hydrological models for green roofs simulation," *Water Science and Technology*, vol. 81, no. 7, pp. 1375–1387, Apr. 2020, doi:10.2166/wst.2020.210. |
| 82. | X. Haowen *et al.*, "Comparing simulations of green roof hydrological processes by SWMM and HYDRUS-1D," *Water Supply*, vol. 20, no. 1, pp. 130–139, Feb. 2020, doi:10.2166/ws.2019.140. |
| 83. | L. Liu, L. Sun, J. Niu, and W. J. Riley, "Modeling Green Roof Potential to Mitigate Urban Flooding in a Chinese City," *Water*, vol. 12, no. 8, p. 2082, Jul. 2020, doi:10.3390/w12082082. |
| 84. | J. Schade, S. Lidelöw, and J. Lönnqvist, "The thermal performance of a green roof on a highly insulated building in a sub-arctic climate," *Energy and Buildings*, vol. 241, p. 110961, Jun. 2021, doi:10.1016/j.enbuild.2021.110961. |
| 85. | K. X. Soulis, N. Ntoulas, P. A. Nektarios, and G. Kargas, "Runoff reduction from extensive green roofs having different substrate depth and plant cover," *Ecological Engineering*, vol. 102, pp. 80–89, May 2017, doi:10.1016/j.ecoleng.2017.01.031. |
| 86. | R. Castiglia Feitosa and S. Wilkinson, "Modelling green roof stormwater response for different soil depths," *Landscape and Urban Planning*, vol. 153, pp. 170–179, Sep. 2016, doi:10.1016/j.landurbplan.2016.05.007. |
| 87. | P. Bevilacqua, D. Mazzeo, and N. Arcuri, "Thermal inertia assessment of an experimental extensive green roof in summer conditions," *Building and Environment*, vol. 131, pp. 264–276, Mar. 2018, doi:10.1016/j.buildenv.2017.11.033. |
| 88. | M. Sněhota, J. Hanzlíková, M. Sobotková, and P. Moravcik, "Water and thermal regime of extensive green roof test beds planted with sedum cuttings and sedum carpets," *Journal of Soils and Sediments*, Sep. 2020, doi:10.1007/s11368-020-02778-x. |
| 89. | M. Maiolo, B. Pirouz, R. Bruno, S. A. Palermo, N. Arcuri, and P. Piro, "The Role of the Extensive Green Roofs on Decreasing Building Energy Consumption in the Mediterranean Climate," *Sustainability*, vol. 12, no. 1, p. 359, Jan. 2020, doi:10.3390/su12010359. |
| 90. | L. Yao, Z. Wu, Y. Wang, S. Sun, W. Wei, and Y. Xu, "Does the spatial location of green roofs affects runoff mitigation in small urbanized catchments?," *Journal of Environmental Management*, vol. 268, p. 110707, Aug. 2020, doi:10.1016/j.jenvman.2020.110707. |
| 91. | E. Cristiano, S. Urru, S. Farris, D. Ruggiu, R. Deidda, and F. Viola, "Analysis of potential benefits on flood mitigation of a CAM green roof in Mediterranean urban areas," *Building and Environment*, vol. 183, p. 107179, Oct. 2020, doi:10.1016/j.buildenv.2020.107179. |
| 92. | Y. He, H. Yu, A. Ozaki, and N. Dong, "Thermal and energy performance of green roof and cool roof: A comparison study in Shanghai area," *Journal of Cleaner Production*, vol. 267, p. 122205, Sep. 2020, doi:10.1016/j.jclepro.2020.122205. |
| 93. | M. Ebadati and M. A. Ehyaei, "Reduction of energy consumption in residential buildings with green roofs in three different climates of Iran," *Advances in Building Energy Research*, vol. 14, no. 1, pp. 66–93, Jan. 2020, doi:10.1080/17512549.2018.1489894. |
| 94. | S. Huang, A. Garg, G. Mei, D. Huang, R. B. Chandra, and S. G. Sadasiv, "Experimental study on the hydrological performance of green roofs in the application of novel biochar," *Hydrological Processes*, vol. 34, no. 23, pp. 4512–4525, Nov. 2020, doi:10.1002/hyp.13881. |

**Table A2.** *Cont.*

| | |
|---|---|
| 95. | A. Ragab and A. Abdelrady, "Impact of Green Roofs on Energy Demand for Cooling in Egyptian Buildings," *Sustainability*, vol. 12, no. 14, p. 5729, Jul. 2020, doi:10.3390/su12145729. |
| 96. | V. Skala *et al.*, "Hydrological and thermal regime of a thin green roof system evaluated by physically-based model," *Urban Forestry & Urban Greening*, vol. 48, p. 126582, Feb. 2020, doi:10.1016/j.ufug.2020.126582. |
| 97. | A. Naranjo, A. Colonia, J. Mesa, and A. Maury-Ramírez, "Evaluation of Semi-Intensive Green Roofs with Drainage Layers Made Out of Recycled and Reused Materials," *Coatings*, vol. 10, no. 6, p. 525, May 2020, doi:10.3390/coatings10060525. |
| 98. | L. Cirrincione *et al.*, "Green Roofs as Effective Tools for Improving the Indoor Comfort Levels of Buildings—An Application to a Case Study in Sicily," *Applied Sciences*, vol. 10, no. 3, p. 893, Jan. 2020, doi:10.3390/app10030893. |
| 99. | R. Castiglia Feitosa and S. J. Wilkinson, "Attenuating heat stress through green roof and green wall retrofit," *Building and Environment*, vol. 140, pp. 11–22, Aug. 2018, doi:10.1016/j.buildenv.2018.05.034. |
| 100. | A. Gagliano, F. Nocera, M. Detommaso, and G. Evola, "Thermal Behavior of an Extensive Green Roof: Numerical Simulations and Experimental Investigations," *IJHT*, vol. 34, no. S2, pp. S226–S234, Oct. 2016, doi:10.18280/ijht.34S206. |

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
