# Peer review of "A Roof of Greenery, but a Sky of Unexplored Relations—Meta-Analysis of Factors and Properties That Affect Green Roof Hydrological and Thermal Performances"

_sustainability, doi:10.3390/su131810017_

Round 1
Reviewer 1 Report
The article explores a topic that is currently important. Green roofs are one of the tools for cities to adapt to climate change and a better understanding of this theme is crucial. The analysis is well done, although I have some major comments that need to be taken into account:
- In the abstract, please do not use the abbreviation FPPR but give its full name. In the present form, it is not clear what the authors mean by FPPR because the explanation of the abbreviation is given much later in the manuscript.
- In the introduction section, you mention that existing roofs can be adapted into green roofs (lines 53-57). What are the limitations to this? Can a green roof be installed on all types of roofs? What load-bearing capacity do roofs need to have? Please expand on this theme in your introduction. This is an important aspect and must not be overlooked.
- In line 133 you mention that the designated 25 factors were reduced to just 12. What was the reason for this reduction? Why was this done? How were the factors selected? Please explain this in the manuscript. Please do this also for the properties that you have also restricted.
- The summary section should clearly state the directions for further research that arise from your analysis.
- As an appendix to the manuscript, it is suggested to include a table covering all the publications analysed by you (references). If you do not want to add it as an appendix, you may also consider including it in the main body of the manuscript. Please consider its design/thematic arrangement.
Editing notes:
1) From line 58 to line 87, please change the citation style. Please do not use "surname et al." only according to the requirements of the journal citation in brackets e.g. [1] Same in line 129, 159, 386,387, 419, 437,438, 440,444, 476, 481, 488, 502. Please check the whole manuscript carefully and adjust the citation style according to MDPI requirements.
2) Please format all tables and captions above tables according to journal style. Likewise the captions under the figures.
3) In line 315, the number of the figure you refer to is missing.
4) In line 372 there is an unnecessary dot at the beginning.
Author Response
|
Dear Madam/Sir.
Thank you very much for your time and valid inputs which will enhance the quality of the paper.
• In the abstract, please do not use the abbreviation FPPR but give its full name. In the present form, it is not clear what the authors mean by FPPR because the explanation of the abbreviation is given much later in the manuscript. Sorry for that and edited the abstract (line 24-25).
• In the introduction section, you mention that existing roofs can be adapted into green roofs (lines 53-57). What are the limitations to this? Can a green roof be installed on all types of roofs? What load-bearing capacity do roofs need to have? Please expand on this theme in your introduction. This is an important aspect and must not be overlooked. Thank you for the suggestion. We added lines 57 - 59 and two new references regarding the same. • In line 133 you mention that the designated 25 factors were reduced to just 12. What was the reason for this reduction? Why was this done? How were the factors selected? Please explain this in the manuscript. Please do this also for the properties that you have also restricted. Indeed, it was not clear enough. We edited the text by adding lines 145 - 147 and line 154. We also added the full tables as Appendix A.
• The summary section should clearly state the directions for further research that arise from your analysis. Thank you for the suggestion. We edited the end of the abstract and rewrote partly the conclusion (also as suggested by Reviewer 2).
• As an appendix to the manuscript, it is suggested to include a table covering all the publications analysed by you (references). If you do not want to add it as an appendix, you may also consider including it in the main body of the manuscript. Please consider its design/thematic arrangement. Great idea! We added it as Appendix B
Editing notes: 1) From line 58 to line 87, please change the citation style. Please do not use "surname et al." only according to the requirements of the journal citation in brackets e.g. [1] Same in line 129, 159, 386,387, 419, 437,438, 440,444, 476, 481, 488, 502. Please check the whole manuscript carefully and adjust the citation style according to MDPI requirements. 2) Please format all tables and captions above tables according to journal style. Likewise the captions under the figures. 3) In line 315, the number of the figure you refer to is missing. 4) In line 372 there is an unnecessary dot at the beginning. Sorry again for that. All considered and edited accordingly
Thank you again for your valuable suggestions. Sincerely, Authors |
Reviewer 2 Report
The paper entitled “A roof of greenery, but a sky of unexplored relations – Meta-
analysis of factors & properties that affect green roof hydrological & thermal performances”
investigated factors and properties that influence the different hydrological and thermal performances of green roofs. It is a very interesting "review paper", well-structured which addresses clearly the issues.
The introduction reports some papers but enough to understand the background. Moreover, many of them are not quoted in agreement with the guidelines of the journal (i.e. “In Sweden, Bengtsson et al. (2005) and Villarreal and Bengtsson (2005). Please, you should check the citations and guidelines. In addition, other important papers are missing ( i.e. :https://doi.org/10.18280/ijht.34S206 ) for a complete, comprehensive analysis and understanding.
The conclusions should be extended adding some meaningful results.
Author Response
Dear Madam/Sir.
Thank you very much for your time and valid inputs which will enhance the quality of the paper.
Please find below the response to your suggestions made.
- Moreover, many of them are not quoted in agreement with the guidelines of the journal (i.e., “In Sweden, Bengtsson et al. (2005) and Villarreal and Bengtsson (2005)
Sorry for that. We corrected all the citations in accordance with the guidelines of the journal.
- In addition, other important papers are missing (i.e.: https://doi.org/10.18280/ijht.34S206) for a complete, comprehensive analysis and understanding.
Thank you for the suggestion. Taking your suggestion into account, we added the article and hence updated the figures and result section too accordingly.
- The conclusions should be extended adding some meaningful results.
Thank you for the suggestion. We edited the end of the abstract and rewrote partly the conclusion (also as suggested by Reviewer 1)
Thank you again for your valuable suggestions.
Sincerely,
Authors
Round 2
Reviewer 1 Report
I am very satisfied with the corrections the authors have made to the manuscript. All my comments have been taken into account. The paper is now good. In my opinion, the manuscript is ready for publication. Please just format correctly all references up to number 60 in the reference list! First should be the author's surname then the first name and then all the elements according to the guidelines. Please also look at the guidelines for the font (bold, italic, etc.) in which you write the name of the journal and the year of publication in MDPI because you have not followed the instructions.